# Optical Wireless Link Operated at the Wavelength of 4.0 µm with Commercially Available Interband Cascade Laser

**DOI:** 10.3390/s21124102

**Published:** 2021-06-15

**Authors:** Janusz Mikołajczyk, Robert Weih, Marcin Motyka

**Affiliations:** 1Institute of Optoelectronics, Military University of Technology, S.Kaliskiego 2, 00-908 Warsaw, Poland; 2Nanoplus Nanosystems and Technologies GmbH, Oberer Kirschberg 4, 97218 Gerbrunn, Germany; robert.weih@nanoplus.com; 3Department of Experimental Physics, Faculty of Fundamental Problems of Technology, Wroclaw University of Science and Technology, Wybrzeże Wyspiańskiego 27, 50-370 Wrocław, Poland; marcin.motyka@pwr.edu.pl

**Keywords:** optical wireless communication, interband cascade lasers, optical link, free space optics, MWIR data link

## Abstract

This paper evaluates the key factors influencing the design of optical wireless communication (OWC) systems operating in the mid-infrared range. The performed analysis has shown that working in this spectral “window”, compared to other wavelengths, is more effective in reducing the attenuation of radiation. The main goal was to verify the capabilities of the “on-shelf” interband cascade (IC) laser in the context of OWC system construction, considering its output power, modulation rate, room temperature operation, and integrated structure. For this purpose, a lab model of a data link with IC laser has been developed. Based on its main parameters, the estimation of signal-to-noise power ratio versus data link range was made. That range was about 2 km for a case of low scintillation and relatively low visibility. In the experimental part of the work, the obtained modulation rate was 70 MHz for NRZ (non-return-to-zero) format coding. It is an outstanding result taking into consideration IC laser operated at room temperature.

## 1. Introduction

Nowadays, microwave communication technologies have reached the upper limit of data throughput. As a result, the interest in laser technology in optical wireless communication (OWC) has been renewed. From the practical point of view, this technology applies to devices operating in the range of several atmosphere transmission windows, which are characterized by lower propagation losses. These windows are located around 0.8 μm and 1.55 μm in near-infrared (NIR), from 3.0 to 5.2 μm in mid-infrared (MWIR), and from 8 to 12 μm in long-wavelength infrared (LWIR) [1]. Commercially available OWC systems most often operate in the NIR range in which well-developed optoelectronic technologies exist.

MWIR is a highly transparent optical transmission window in the atmosphere, mainly characterized by a “lack” of water vapor absorption. There is a lower beam distortion due to beam scattering, beam wandering, loss of spatial coherence, or scintillation [2,3]. However, developing such technology requires the MWIR radiation sources with high power, eye-safe, small size and cost, and room temperature (RT) operation. Quantum Cascade Laser (QC laser) and Interband Cascade Laser (IC laser) have great potential as components of novel OWC systems. Some experimental results of using QC lasers to transmit data with the rates from tens of Mb/s to a few Gb/s at wavelengths from 3 µm to 4.8 µm and 77 K or 300 K temperatures are discussed in the literature [4,5,6,7,8,9,10]. However, most of them concern OWC laboratory setups with QC laser structures working at very short distances.

The very good operating parameters of ICLs, such as low threshold currents resulting in low power consumption [11], relatively high output power [12], wide tunability of the emission wavelength [13,14], have found numerous applications in medical diagnostics [15] and environmental protection (monitoring of, e.g., formaldehyde [16]). But only a few tests of the optical link have been described. For example, the laser operated at T = 77 K with an output power of ~ ten mW and a wavelength of 3.0 μm was used to obtain a data rate of 70 Mb/s [8]. A construction of two-mm long lasers, mounted onto an oxygen-free copper holder with temperature-controlled cold finger of the optical cryostat and wire bonded to 50 Ω RF microstrip line, was also described in [4]. The results were obtained using unique designed lasing structures operated at 77 K with a pulse power of 10 mW.

Although the IC laser technology is still developing, the research area of its application in wireless communications is still unexplored. The main goal of the performed works was to define the state-of-the-art of this technology enabling transmission of data signals. It should also be emphasized that, to the best of the acquired knowledge, this is the first demonstration of the data link using the “on-shelf” IC laser operated at room temperature at the wavelength of ~4 µm. The main virtues of this link compared with some OWC setups described in the literature are listed in Table 1.

## 2. Materials and Methods

### 2.1. Calculations

OWC system transmits the data using optical radiation; therefore, its main elements are a transmitter (laser source) and a receiver (photodetector). Data link range is determined by the required ratio of the received signal power to the noise power. Power of the detected signal $P_{d}$ is determined by [17]:(1)$$P_{d}=P_{0}Ge^{-\gamma \left(\lambda \right)L},$$
where $P_{0}$ is the laser light power, γ(λ) is an extinction coefficient [1/km], L is a link distance, and G is a gain. In such a case, the gain is defined by geometrical relation:(2)$$G=\frac{D_{d}^{2}}{{\left(D_{0}+\theta_{div}L\right)}^{2}},$$
where $\theta_{div}$ is the beam divergence, $D_{0}$ and $D_{d}$ are the aperture diameters of the transmitter and optical receiver components. The coefficient γ(λ)  accounts for the beam attenuation caused by the absorption and scattering because of the light interaction with air molecules and aerosol particles [18,19]. The absorption attenuation is minimized by working in the spectral range of atmospheric windows, so in practice, scattering becomes one of the “bottlenecks” of the entire OWC system operation. Its character depends on the size distribution of the scattering particles in comparison to the radiation wavelength. For MWIR radiation, the scattering effects are described using the Mie theory. The light losses are estimated using “visibility—Vis” combined with weather conditions [20,21]. This attenuation can be transferred into MWIR range using a function described empirically by Kruse [22]:(3)$$\beta \left(\lambda \right)=\frac{3.91}{Vis}{\left(\frac{\lambda }{\lambda_{0}}\right)}^{-q},$$
where *λ*_0_ = 550 nm is the reference wavelength, and q is the particle size distribution coefficient. Kruse model describes this coefficient as 1.6 for visibility better than 50 km, 1.3 for visibility in the range of 6 km ÷ 50 km and equals to 0.585 *Vis*^1/3^ for visibility below 6 km. However, the Kruse model is assumed to be good enough for visibility better than 1 km. To analyze other propagation conditions, one would need to utilize other models of scattering losses, which can be found in the literature [23]. In the case of fog attenuation, two attenuation functions are described [24]. Because the advection fog is characterized by a linear attenuation wavelength dependence and the radiation fog by a quadratic one, higher light losses are observed for the radiation fog. There is also a third scattering effect caused by rain, snow, and dust. Some models are also available to determine their respective attenuation [25]. Generally, these particles are larger than the operating wavelength, and scattering becomes wavelength independent [26,27,28]. The analytical calculations of the extinction coefficient for different weather conditions for 1.55 µm and 4.0 µm wavelengths were performed using the PcModwin 6.0 software (Ontar, North Andover, MA, USA) (Table 2). The first wavelength corresponds to the operation spectrum of now available commercial OWC systems, and the second one to the spectral region of ICL emission.

For “clear” air and high visibility, the light attenuation is mainly determined by the absorption. In this case, the atmosphere is more transparent for MWIR radiation (Figure 1) [29].

For rain, the same beam attenuation is noticed for the considered wavelengths, and there are also no significant differences for very low visibility (below 500 m). However, if the visibility is increased, the propagation conditions of light improve again for longer wavelengths.

Noise in OWC systems is mainly generated in the receiver. There are different sources of noise in both photodetector and readout electronics (amplifiers). Its detectivity usually describes the detector noise [30]:(4)$$D^{*}=\frac{A_{d}\sqrt{∆f}}{NEP},$$
where $A_{d}$ is the active detector surface, ∆*f* is the bandwidth, and NEP is the noise equivalent power. The total output noise power is equal to:(5)$$P_{t}=P_{n}K+P_{a},$$
where $P_{n}$ is the input noise power (photodetector noise), K is the amplifier gain, and $P_{a}$ is the amplifier noise. There is observed dynamic progress in infrared photodetectors technology allows to obtain very high detectivity [31]. Nowadays, the optimization of readout electronics also ensures ultra-low noise levels below the photodetector one [32].

Optical link performances are very often modified by air turbulence [33]. This phenomenon is based on a random variation of the refractive index generated by optical cells (eddies) of different sizes formed during fluctuations of temperature and pressure in the atmosphere. Due to the described phenomena, the propagating beam is disturbed, and random changes of the received signal are observed. As a result, the effects of beam wandering, beam spreading, or scintillation can be registered. Simplifying, it causes random changes in signal-to-noise power ratio (SNR). This effect is described by the scintillation index corresponding to the normalized fluctuation of the signal [34]:(6)$$\sigma_{I}^{2}=\frac{\langle I^{2}\rangle -{\langle I\rangle }^{2}}{{\langle I\rangle }^{2}},$$
where I is the light intensity on the detector surface, and 〈I〉 is its mean value. The signal variance is determined by a refractive index structure parameter (*C*_n_^2^—index), link range, and a light wavelength. The *C*_n_^2^ parameter defines the level of turbulence. Theoretically, the operation in longer wavelengths allows obtaining better transmission during a moderate level of turbulence. For strong turbulence, the same attenuation for OWC wavelengths is observed. The effects of scintillation influence are challenging to predict, considering the complex dependence on temperature and pressure distribution, humidity, altitude, surface character, and more. Nowadays, several mathematical models have already been developed, which, with some approximation, provide the required accuracy to define this phenomenon. Still, each of them has some limitations (level of turbulence, altitude, and distance). To summarizing, the effect of scintillation cannot be eliminated but can be minimized [35].

The OWC system availability depends on weather conditions and parameters, such as transmitted optical power, beam divergence, receiver sensitivity, or link path distance [36].

To visualize the influence of scintillation on the OWC system operating at two attractive wavelengths (1.5 µm and 4 µm), a mathematical model of such data link based on the analytical dependence described in [37,38] was used. In Figure 2, data link ranges are shown for OWC systems providing SNR = 10 at different turbulence levels (*C_n_*^2^ values). The parameters of these systems and beam propagation conditions were the same (a laser power of 600 mW, the four-inches diameter of receiving optics, beam divergence of 1 mrad, 3 × 10^9^ cm√Hz/W-detector detectivity with 1 × 1 mm^2^ active surface, 700 MHz-signal bandwidth, and an extinction coefficient of 0.25 km^−1^). The estimated data range is above 1 km for both wavelengths with the advantage of 4 µm, and it increases commonly if *C_n_*^2^ parameter is lower than 10^−15^ m^−2/3^ (weak turbulence).

The practical limitation of an OWC system is also a pointing error generated by moving its transceivers from the line of sight. This misalignment results from building sway and mechanical vibration caused by thermal expansions, strong wind, and weak earthquakes [39]. This effect decreases the average received signal and, consequently, increases the bit-error rate (BER). However, it is a common limitation for OWC systems.

### 2.2. Experiment

In the designed OWC lab setup, an ICL from Nanoplus Nanosystems and Technologies GmbH was used. The active region of this laser comprises six active stages that are embedded in 430 nm thick GaSb separate confinement layers on both sides. The epi side up mounted laser chip is not optimized for high power operation, has a cavity length of 900 µm and a ridge width of 5.1 µm to ensure lasing on the fundamental lateral mode. The back facet is coated with a high reflectivity coating, whereas the front facet is just covered with a thin dielectric layer to protect the emitting facet. The laser structure was mounted into the “TO-66” semiconductor package with a temperature Peltier module and a temperature sensor. In Figure 3, the spectral characteristic of the radiation at 4 µm was shown (panel **a**) together with light-current-voltage (*LIV*) characteristics (panel **b**).

The schematic diagram of the lab setup is presented in Figure 4. In the OWC transmitter, a programmable current pulse driver model MAX3967 supplies the ICL. This driver is dedicated to controlling fast LED sources, but the range of generated current and voltage signals effectively supplies such lasers. Also, a programmable biasing voltage from 400 mV to 925 mV was implemented. The laser temperature of 25 °C was stabilized using the Peltier temperature controller model Arroyo 5310. The laser beam was collimated using an off-axis mirror with a diameter of three inches and a focal length of two inches.

The model of the OWC receiver consisted of a four-inches off-axis parabolic mirror, InAsSb photodetector (VIGO System S.A.), and readout electronics were used. The photodetector parameters are listed in Table 3.

The detector electronics has a cascade configuration with the first stage based on a low noise transimpedance (TIA) and a voltage amplifier. In both, low-noise, high bandwidth opamps were used (OPA847 model). The designed electronics’ current noise density and signal bandwidth were 20 pA/√Hz and 140 MHz, respectively. These characteristics define the theoretical limits of the data link range (noise power level of 532 nW) and its transfer rate (~180 Mb/s) for NRZ format coding.

For the described configuration, the data range for different weather conditions was calculated. Finally, the laboratory tests of the OWC link were also performed using both RZ and NRZ coding formats. A pattern signal generator model 12,000 Picosecond controlled the laser pulses. The output signal from the receiver was registered and analyzed using an oscilloscope model MSO 6 Tektronix with a built-in eye diagram toolbox.

## 3. Results

### 3.1. Calculations

Estimation of the data range of the configured link was performed using the mathematical model of the OWC system. The main task was to determine the changes in the signal-to-noise power ratio in function of the distance considering parameters of its components for different atmospheric conditions defined by the extinction coefficient and the scintillation level (“weak” $C_{n}^{2}\;$= 10^−15^ m^−2/3^, “moderate” $C_{n}^{2}$ = 10^−14^ m^−2/3^ and “strong” $C_{n}^{2}\;$= 10^−13^ m^−2/3^). Calculations of SNR allow relating the obtained results to different modulation standards. Additionally, some analyses of using higher power ICLs to increase the operating range were also carried out. In Figure 5, calculated SNR changes for different weather condition scenarios and laser power were presented.

A strong influence of scintillation at short link distances (Figure 5a) was observed. Increasing the scintillation level causes a dynamic decrease in the SNR value, especially at a distance of less than 500 m. For the configured OWC system, the “last mile” data transmission is achieved with the SNR = 10 if the structure parameter does not exceed 10^−14^ m^−2/3^, assuming good other weather conditions. This level corresponds to the turbulence that can occur during the operation of terrestrial OWC systems, for example, placed near the ground. Such systems are installed considering, e.g., distance and type of surface, wind, opening area, sun shading. For different weather conditions causing light scattering, a significant range limitation occurs below the visibility of 1 km (Figure 5b). It corresponds to the formation of haze. If the weather conditions are excellent, the range is mainly determined by geometric loss and radiation absorption in the air. Increasing the power of the laser pulses does not directly increase the registered SNR value at short distances, where scintillation is the main limitation. However, higher power is required at longer distances to compensate for the radiation losses caused by the extinction coefficient and beam divergence (Figure 5c). The described OWC configuration provides to obtain SNR = 10 at the distance of about 2 km for weather conditions defined by a structure parameter of 10^−15^ m^−2/3^ and visibility of 1 km. It should also be noticed that the four times increase in the laser power extends this distance only by 0.5 km for these conditions.

### 3.2. Experiment

In the first step of the experiments, the test of laser pulse generation with RZ format coding was performed. Figure 6 presents a 16-bit information frame in which 6 bits of “1” are written. The amplitude fluctuations of the pulses were observed depending on the quantity of “1”-bit. For a higher number of “1”, these fluctuations were decreased, and a significant reduction in pulse amplitude was also noticed.

One bit of “1” was placed in the frame to determine the maximum pulse frequency for RZ format coding. The maximum bit frequency equals 155 MHz. However, in this case, both laser pulse rise time and receiver signal bandwidth are the main limiting factors of the modulation rate. It can be observed on the shapes of the registered laser pulses with different time duration—for the shortest pulses, the pulse amplitude becomes twice lower when the pulse is reduced from 20 ns down to 5 ns.

Further, a data transmission test of NRZ format coding was also carried out. The laser was driven using PRBS (pseudo-random bit sequence) signal of 12-bits frame with five bits of “1” (001101000110) using the same generator. For the developed OWC system, the maximum frequency for which the eye diagram was still recorded was 70 MHz (Figure 7a). An observed bath tube diagram with a bit error rate (BER) of 10^−9^ was also presented in Figure 7b. For the frequencies above 78 MHz, the eye diagram was closed from time to time; therefore, for the described OWC configuration with “on-shelf” ICL, the maximum modulation frequency for NRZ format coding was defined at 70 MHz.

## 4. Conclusions

The benefits of optical wireless communication systems make this technology very promising for novel wireless communications. However, some limitations have been defined by the light losses of the free space propagation determined by air compounds (absorption), weather conditions (scattering), or air turbulence (scintillation). Nowadays, there are several available systems operated in the near-infrared spectrum. However, compared to the commercially 1.55 µm-data link, the described analysis determinates a “better transparency” of 4.0 µm optical link for different weather conditions. However, it was also shown that higher laser power could provide a more extended data link but is not so practical for compensating the scintillation effects at short distances.

The practical study of interband cascade laser application in optical data transmission is different from the previous approaches because it is the first link setup in which an “on-shelf” interband cascade laser operated at room temperature was tested. For return-to-zero format coding, the influence of the “data frame” shape on pulse fluctuations and amplitudes was observed. It is crucial considering the dynamic response of the receiving units. The maximum modulation rates were 155 MHz and 70 MHz for return-to-zero and non-return-to-zero format coding and defined a practical validation of the interband cascade laser-based technology level. The calculated data link range of the described setup for low-level scintillations and low visibility was about 2 km. These results indicate opportunities for future optical wireless communications development.

## Figures and Tables

**Figure 1 sensors-21-04102-f001:**
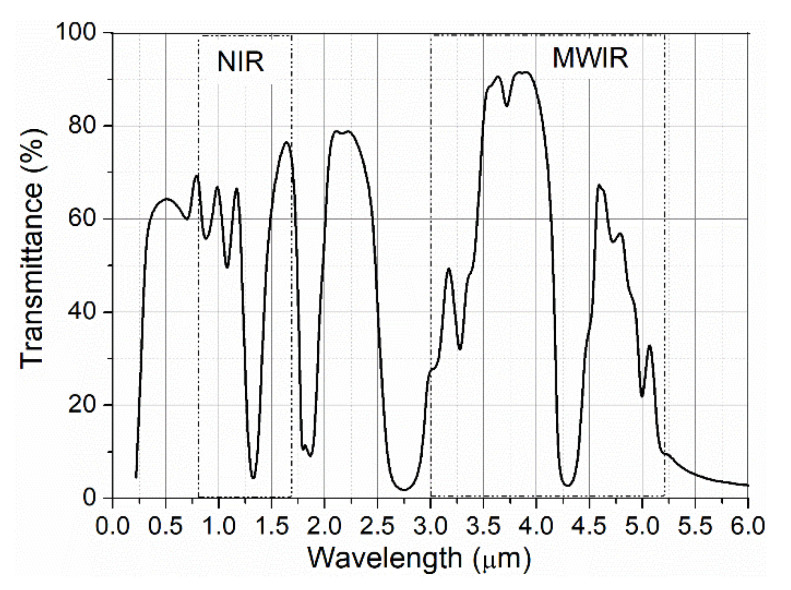
Spectra of atmospheric transmittance.

**Figure 2 sensors-21-04102-f002:**
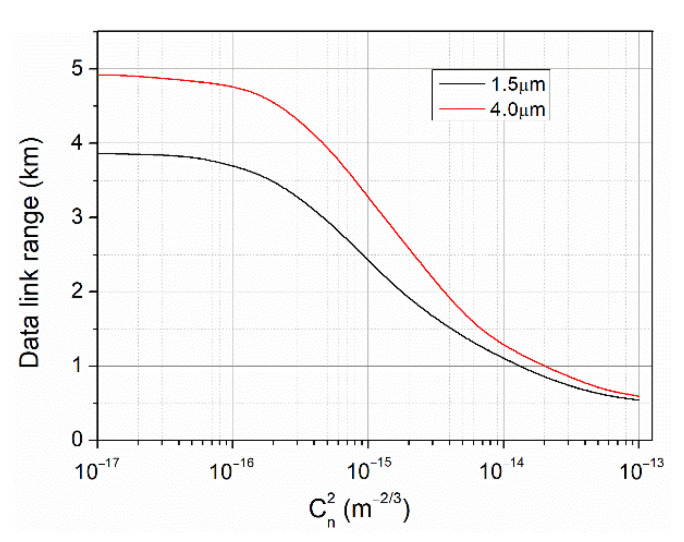
OWC link range vs. scintillation level for two interesting wavelengths.

**Figure 3 sensors-21-04102-f003:**
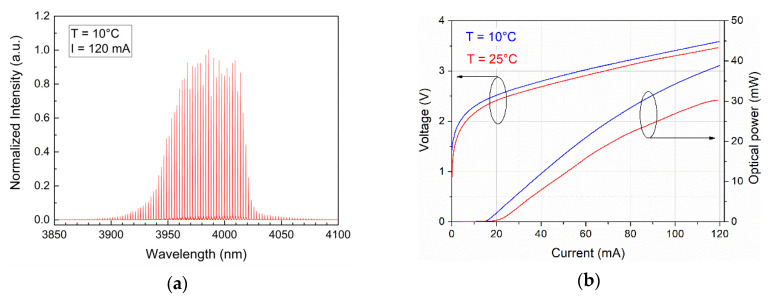
(**a**) ICL radiation spectrum and (**b**) *LIV* characteristics.

**Figure 4 sensors-21-04102-f004:**
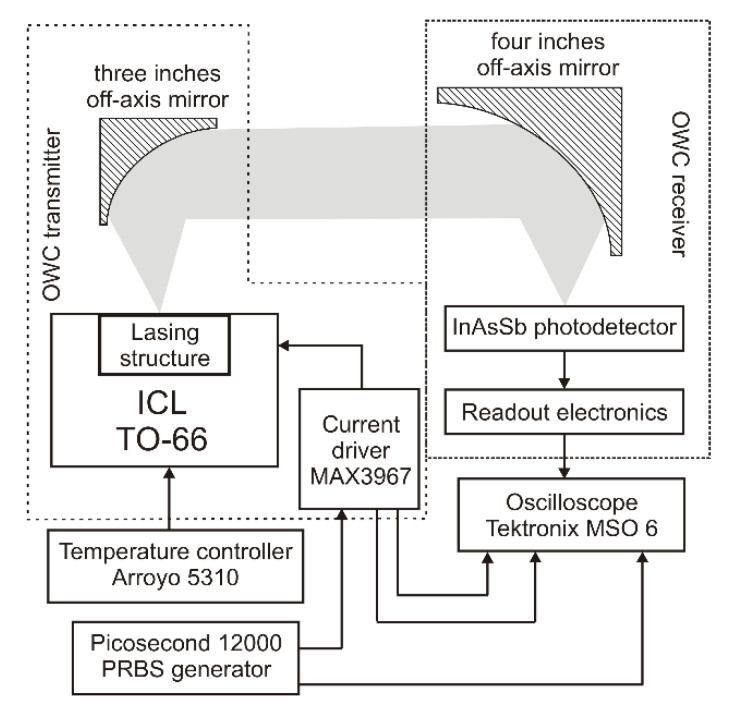
Scheme diagram of the OWC lab setup.

**Figure 5 sensors-21-04102-f005:**
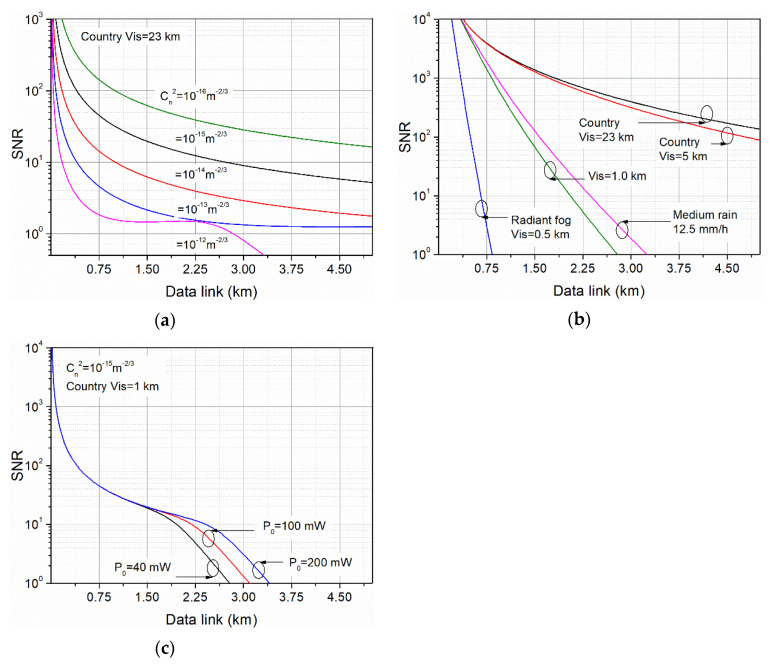
OWC data range for different propagation conditions determined by (**a**) scintillation, (**b**) visibility, and (**c**) pulse laser power.

**Figure 6 sensors-21-04102-f006:**
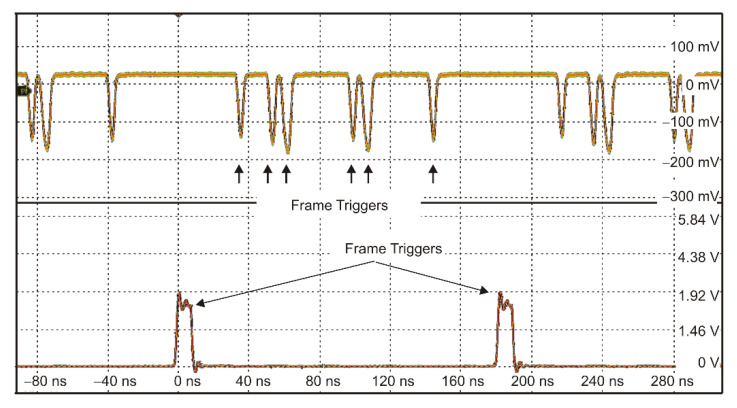
A frame of pulses with 6 bits of “1”.

**Figure 7 sensors-21-04102-f007:**
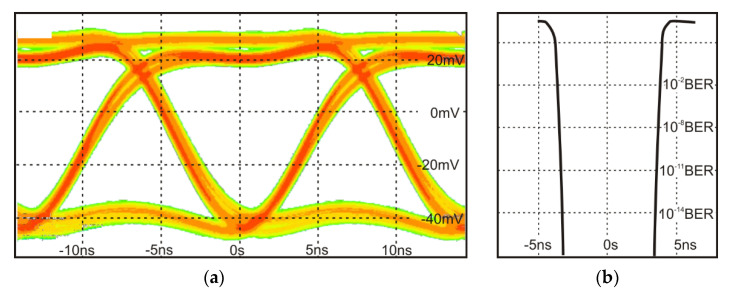
(**a**) Eye diagram and (**b**) bathtube of NRZ format coding laser pulses at the maximal frequency of 70 MHz.

**Table 1 sensors-21-04102-t001:** Examples of the OWC system operated in the MWIR spectrum.

Wavelength	Laser	Data Rate	Modulation *	Distance	Temperature	Year
3.0 μm	ICL	70 Mb/s	*NRZ-OOK*	1 m	77 K	2009 [4]
4.7 μm	QCL	40 MHz	Analog	2.5 m	293 K	2015 [5]
4.65 μm	QCL	3 Gb/s	*NRZ-OOK*, *PAM-4*, *PAM-8*	5 cm	293 K	2017 [6]
4.65 μm	QCL	4 Gb/s	*PAM-4*, *DMT*	cm	293 K	2017 [9]
4.8 μm	QCL	10 MHz	Analog	m	293 K	2020 [7]
3.0 μm	ICL	70 Mb/s	*OOK*	cm	77 K	2010 [8]
4.0 μm	ICL	70 Mb/s 155 Mb/s	*NRZ-OOK* *RZ-OOK*	2 m	298 K	This work

* Modulation formats: *RZ*—return-to-zero, *OOK*—on-off keying, *PAM*—pulse amplitude modulation, *DMT*—discrete multitone.

**Table 2 sensors-21-04102-t002:** The extinction coefficient γ [km^−1^] of two wavelengths *λ* and for different weather conditions from PcModwin 6.0.

λ	Aerosol Model *, Vis [km]								
	*A*	*C* 23 km	*C* 5 km	*T* 5 km	*M* 23 km	*D* 23 km	*F* 0.5 km	*R* 12.5 mm/h	*H* 2 mm/h
1.5 μm	0.20	0.24	0.40	0.43	0.31	0.22	8.92	2.08	0.86
4.0 μm	0.05	0.06	0.13	0.15	0.12	0.07	10.58	1.88	0.65

**A*—only absorption, *C*—country, *T*—town, *M*—maritime, *D*—desert, *F*—fog, *R*—rain, *H*—haze.

**Table 3 sensors-21-04102-t003:** Detector parameters and spectral response.

Parameter	Value	Spectral Characteristics
Active area	1 × 1 mm^2^	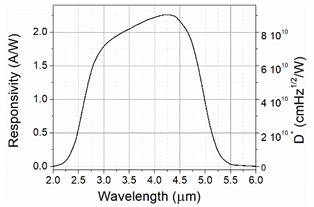
Detector resistance	60 kΩ	
Current noise density	2.4 pA/√Hz	
Responsivity	2.25 A/W	

## Data Availability

Not applicable.

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
