# Peer review of "Optical Wireless Link Operated at the Wavelength of 4.0 µm with Commercially Available Interband Cascade Laser"

_sensors, 2021, doi:10.3390/s21124102_

Round 1
Reviewer 1 Report
In the manuscript, the authors investigate the potentials of a commercial laser operating in the mid-infrared spectral region for studying the design of optical wireless communication systems. A lab set-up with the commercial laser under test has been used to simulate the impact of different real environmental conditions (fog, rain, scintillation, etc.) on the optical wireless link performance. In general, the paper is well organized and technically sounds. However, the following issues and questions should be addressed by the authors:
- In the manuscript, it has been underlined several times and finally concluded that a laser working at room temperature is one of the main aspects of the presented work. However, as shown in Fig.2, the used laser was characterized at 10 degrees of Celsius. Why? It might be better to replace these results with the ones that are obtained at room temperature.
- In the abstract and conclusions, the authors mention that the performed/described analysis shows the mid-infrared is a better window than the near-infrared. These sentences are inaccurate since a limited comparison of the wavelength regions is provided in the manuscript. Table 1 provides the data taken from the software and Figure 1 compares 1.5 µm and 4 µm wavelengths only in terms of scintillation.
- The texts in Fig. 3a should be corrected. Most of them indicate the same value of Cn, and the font should be bigger for being more readable.
- A sketch of the setup could be provided for clarification. The measurement method used to mimic the parameters Cn and Vis is not clearly explained. Since the experimental work is done in the lab, one can assume that the only variable to chance would be the attenuation. However, there is not sufficient information provided in the manuscript about how it is done.
Author Response
Dear Reviewer,
At first, I would like to thank you for your suggestions and opinion. They are very accurate and significantly increase the research level of my work. The response and changes are listed in the attached file.

Reviewer 2 Report
This paper presents a optical wireless communication setup in which the capabilities of on-shelf inter-band cascade laser was tested at room temperature. The paper is well-written. However, following suggestions can improve its quality.
1. The literature review is very limited. Include more relevant research article to clearly present the background of proposed problem and research gap.
2. The figure of actual experimental setup is missing. It should be included to enhance the readability of the paper.
3. Please avoid abbreviations in the conclusion section.
4. Please proofread the paper for correction of all typos . Line 31, "from 8 to12". Line 79, "range of 6 km÷50 km". Line 162, "and a voltage amplifiers."
Author Response

(The authors gave the same response as above.)

Reviewer 3 Report
Dear Editor,
I am writing back my review on research article named as "Optical wireless link operated at the wavelength of 4.0 μm with commercially available interband cascade laser" exploring opportunities for optical wireless communications at mid-IR wavebands. Authors evaluated key metrics that have a major impact on the design of such communications. The research work presented herein combines theory with experiments, verifying abilities of interband cascade laser.
This research article under review is reasonably organized, prepared and well-written. This overall readability is high. Indeed, the topic of wireless optical communication as well as operation in the mid-IR wavelengths is a hot topic, with rapid progress, this work can be potentially interesting for a wide community. Adopted methodology is clear and wisely chosen, results presentation and discussion is solid as well. Provided figure are clear. However, prior to the final decision on this manuscript, in Reviewer's opinion, this manuscript should be improved. For that task, below are listed my comments.
(1) For Table 1, please re-adjust the vertical descriptions (second row, all columns) into the horizontal one.
(2) For the equations that are not explicitly derived by Authors, I strongly recommend to put respective references, from which these equation were taken.
(3) Please provide an explanation for an extraordinary high responsivity of 2.25 A/W. How this responsivity was obtained?
(4) Please comment on link losses in more detail.
(5) Please comment on the main limitations and discuss key steps to overcome them.
(6) Manuscript would benefit from a benchmark table, comparing this work with state-of-the-art solutions/results.
(7) Under a perfect circumstances, i.e. ideal case scenario, what would be the maximum attainable bandwidth, and thus the maximum achievable speed.
Author Response

(The authors gave the same response as above.)

Round 2
Reviewer 1 Report
After the updates and corrections that have been made, the quality of the presentation of the presented work in the manuscript has been improved. Therefore, according to the reviewer's opinion, the manuscript can be published in its present form.
Reviewer 2 Report
The reviewer is satisfied with the modifications.